# The Impact of the COVID-19 Pandemic on Hospital Services for Patients with Cardiac Diseases: A Scoping Review

**DOI:** 10.3390/ijerph19063172

**Published:** 2022-03-08

**Authors:** Mats de Lange, Ana Sofia Carvalho, Óscar Brito Fernandes, Hester Lingsma, Niek Klazinga, Dionne Kringos

**Affiliations:** 1Department of Public and Occupational Health, Amsterdam Public Health Research Institute, Amsterdam UMC, University of Amsterdam, Meibergdreef 9, 1105 AZ Amsterdam, The Netherlands; m.delange1@amsterdamumc.nl (M.d.L.); o.r.britofernandes@amsterdamumc.nl (Ó.B.F.); n.s.klazinga@amsterdamumc.nl (N.K.); d.s.kringos@amsterdamumc.nl (D.K.); 2Department of Public Health, Erasmus University Medical Center, Postbus, 3000 CA Rotterdam, The Netherlands; h.lingsma@erasmusmc.nl; 3Department of Health Economics, Corvinus University of Budapest, Fővám tér 8, H-1093 Budapest, Hungary

**Keywords:** acute coronary syndrome, cardiovascular diseases, quality of health care, performance indicator, continuity of patient care, COVID-19

## Abstract

This study aims to assess the impact of the COVID-19 pandemic on hospital cardiac care, as assessed by performance indicators. Scoping review methodology: performance indicators were extracted to inform on changes in care during January–June 2020. Database searches yielded 6277 articles, of which 838 met the inclusion criteria. After full-text screening, 94 articles were included and 1637 indicators were retrieved. Most of the indicators that provided information on changes in the number of admissions (n = 118, 88%) signaled a decrease in admissions; 88% (n = 15) of the indicators showed patients’ delayed presentation and 40% (n = 54) showed patients in a worse clinical condition. A reduction in diagnostic and treatment procedures was signaled by 95% (n = 18) and 81% (n = 64) of the indicators, respectively. Length of stay decreased in 58% (n = 21) of the indicators, acute coronary syndromes treatment times increased in 61% (n = 65) of the indicators, and outpatient activity decreased in 94% (n = 17) of the indicators related to outpatient care. Telehealth utilization increased in 100% (n = 6). Outcomes worsened in 40% (n = 35) of the indicators, and mortality rates increased in 52% (n = 31). All phases of the pathway were affected. This information could support the planning of care during the ongoing pandemic and in future events.

## 1. Introduction

As of 12 January 2022, the virus causing COVID-19 disease, SARS-CoV-2, has infected 313 million people globally and caused 5.5 million deaths [1,2,3]. The use of health resources for non-COVID care was minimized and elective procedures and appointments were postponed since the beginning of 2020 [4,5]. The impact of COVID-19 on health care services for patients with non-communicable diseases (NCDs) seems to be severe and involves multiple clinical areas [6,7,8,9,10,11,12].

According to the World Health Organization, cardiovascular diseases, which include cardiac diseases, are the leading cause of death globally [13]. Before the pandemic, a decelerating pace in the improvement of cardiovascular disease mortality was already identified as a major contributor to the slowdown of life expectancy gains in several Organization for Economic Co-operation and Development (OECD) member countries [14]. The continuity of care for patients with cardiac diseases during a crisis is a major concern among healthcare providers [15,16], who seek to strengthen referrals and care pathways and establish and maintain novel models of care. Given the burden of cardiac diseases in healthcare systems globally [13,14], attention to the impact of the COVID-19 pandemic on cardiology patients is justified. Previous studies noticed a decrease in patients presenting with (acute) cardiac conditions and mentioned delays in those patients who eventually did present for hospital care [17,18,19], which could lead to worse clinical outcomes [20].

In a relatively short period, large amounts of scientific evidence have become available on the impact of the pandemic on patients with cardiac diseases in varying countries. This literature is scattered, which makes it difficult to draw conclusions regarding the impact on cardiac care delivery and the use of data to steer health care service delivery improvements during the current pandemic and future public health crises. Hence, reviewing the existing literature that quantifies the magnitude of this impact during the first half of 2020 helps to systematize, synthesize, and consolidate the evidence and to formulate recommendations for policy and practice.

This scoping review is part of a larger project that focuses on performance indicators used for a range of clinical areas during the COVID-19 pandemic. In this study, we aim to assess the impact of the pandemic on hospital care for cardiac patients in OECD countries, focusing on the different phases of the hospital cardiac care pathway (admission, diagnosis, treatment, outpatient care, outcomes).

## 2. Materials and Methods

We pursued a scoping review methodology, considering the extensive number of scientific articles, their heterogeneous character, and the lack of a clear and structured overview of the large sums of literature that have become available on the impact of the pandemic on patients with cardiac conditions. A scoping review can be used to examine emerging evidence and give an overview of the literature and studies available on a certain theme [21]. We followed the methodological framework developed by Arksey and O’Malley [22], further developed by Levac et al. [23], and the PRISMA extension for scoping reviews to report our methodology [24] (Appendix A).

### 2.1. Search Strategy

The MEDLINE and Embase databases were selected to search for this scoping review, as we considered them to sufficiently cover the literature related to the delivery of health care services. Pilot searches were conducted to identify a list of suitable search terms. A medical research librarian was consulted to improve the search strategy and adapt it to both databases. The final search strategy included the following key terms and synonyms: COVID-19, pandemic, non-communicable disease, chronic disease, performance indicator, healthcare quality, healthcare utilization, healthcare delivery, and other closely related terms. The full search strategy for Embase and MEDLINE can be found in Appendix A. The comprehensive search was conducted by the research librarian on 17 March 2021. No limitations were set regarding language or year of publication. Duplicates were removed using EndNote software. Additional articles of relevance were added by hand-searching the reference lists of the included studies.

### 2.2. Study Selection

The following inclusion criteria were set: (1) studies using empirical data on the use of health services; (2) studies describing health outcomes and/or performance indicators during the COVID-19 pandemic; (3) studies that are presented as original journal articles using quantitative or qualitative methods, such as cohort studies, case-control, cross-sectional designs, case reports, systematic reviews, surveys, and meta-analyses. The following exclusion criteria were set: (1) non-primary studies (such as editorials and commentaries); (2) prediction models; (3) clinical case reports; (4) diseases management or health services organization guidelines; (5) studies about the impact on healthcare workers, patients diagnosed with COVID-19, children, or pregnant women; (6) studies primarily performed in non-OECD countries; (7) articles from which only an abstract was available.

### 2.3. Methods of Selection

An initial screening of the retrieved studies based on title and abstract was performed independently by two researchers (ASC, OBF) using Rayyan [25]. Studies considered relevant after this phase were exported to a spreadsheet to support full-text screening, which was performed independently by three researchers (MdL, ASC, OBF). For this study, only articles on cardiac diseases were analysed. The reason for the exclusion of articles was recorded at this point. In case of doubt, the other co-authors were consulted, and a decision was made.

### 2.4. Data Extraction and Charting

Data extracted from the included articles were collated in a spreadsheet informed by a pilot on 15 studies (Appendix A). Data extraction was performed independently by three researchers (MdL, ASC, OBF). Extracted data included information on generic and methodological aspects of the article (e.g., authors, title, setting) and information about the indicators reported (e.g., indicator title, and data inclusion/exclusion considerations). For every indicator, we identified the trend reported in the articles (increase/decrease/stable).

### 2.5. Synthesis of the Results

Indicators were grouped and categorised according to the different phases of the hospital cardiac care pathway by MdL, followed by a review conducted by ASC (Appendix A). The percentages of indicators showing a decreasing, increasing, or stable trend were computed for each category. Results are presented in line with the hospital cardiac care pathway.

## 3. Results

Database searches yielded 6277 articles. Of these, 838 articles focusing on non-communicable diseases met the inclusion criteria. After full-text screening of 117 articles focused on cardiac hospital care, a total of 94 articles were included in this review (Figure 1). Twenty-three full-text studies were excluded.

### 3.1. General Characteristics of the Included Articles

The included articles reported on 109 different countries (Figure 2). Eighty-six articles provided information on one country only. Eight articles involved multiple countries, of which seven also included non-OECD countries. Most articles reported on Italy (n = 20, 21%), followed by the United Kingdom (n = 17, 18%) and Germany (n = 14, 15%).

Most of the studies used a retrospective cohort design (n = 66, 70%), while other studies used a prospective cohort (n = 13, 14%) or survey design (n = 12, 13%). The included studies used three different time periods when comparing pandemic versus pre-pandemic indicators. Most studies compared a COVID-19-affected period in 2020 to the same period in the previous year (n = 54, 56%). Other studies (n = 34, 36%) compared the COVID-19 affected period to a period immediately before. Lastly, a COVID-19-affected period was compared to the average of the same period in several previous years (n = 27, 29%). The general characteristics for each article can be found in detail in Appendix A.

### 3.2. Impact of COVID-19 on the Hospital Cardiac Care Pathway

The grouped and categorized indicators were collated according to the different phases of the hospital cardiac care pathway (Figure 3) to visualize their combined trends.

#### 3.2.1. Admission

A total of 287 indicators regarding the ‘admission’ phase of the hospital cardiac care pathway were identified. These indicators were grouped into three separate categories: ‘admission to care’, ‘delayed presentation’, and ‘patients’ clinical severity at presentation’.

Admission to care

Regarding the number of admissions for cardiac diseases to hospital services, 134 indicators were identified from 49 articles [10,26,27,28,29,30,31,32,33,34,35,36,37,38,39,40,41,42,43,44,45,46,47,48,49,50,51,52,53,54,55,56,57,58,59,60,61,62,63,64,65,66,67,68,69,70,71,72,73]. Most of the indicators concerned patients with acute coronary syndrome (ACS), heart failure, or cardiac arrythmia (Figure 4). Of these indicators, 118 (88%) reported a decrease in the number of admissions, compared to a non-COVID-19 period, 13 indicators (9.7%) reported a stable number of admissions, whereas 3 indicators (2.2%) reported an increase in admission numbers.

Delayed presentation

Regarding the patients’ timing of presentation to hospital services, 17 indicators were identified from 12 articles [36,43,48,61,71,74,75,76,77,78,79,80]. All the indicators in this category reported on patients with ACS. Most indicators defined delayed presentation as more than 12 h after symptom onset. Of these indicators, 15 (88%) reported an increase in the number of patients with a delayed presentation to hospital services for cardiac care compared to a non-COVID-19 period. One indicator (5.9%) reported a stable number of patients with a delayed presentation and another indicator a decrease.

Patients’ clinical severity

Regarding the patients’ clinical severity, 136 indicators were identified from 38 articles [19,27,31,36,37,39,41,43,46,48,50,52,54,59,61,62,63,66,69,71,73,75,76,79,81,82,83,84,85,86,87,88,89,90,91,92,93,94]. Most of the indicators were markers for patients with ACS (Figure 4), such as left ventricular ejection fraction, Killip class, or biomarkers at admission. Other indicators reported on heart failure and cardiac surgery. Of these indicators, 54 (40%) reported a worse clinical condition at admission when compared with patients during the COVID-19 period, 72 indicators (53%) reported a stable clinical condition, and 10 indicators (7.4%) reported a better clinical condition.

#### 3.2.2. Diagnosis

A total of 19 indicators regarding the ‘diagnosis’ phase of the hospital cardiac care pathway were identified from 11 articles [42,51,54,58,64,71,73,82,95,96,97]. All the indicators reported on the same category: the number of diagnostic procedures, including transthoracic and transoesophageal echocardiograms, non-invasive ischemia tests, and coronary angiographies (Figure 4). Of these indicators, 18 (95%) reported a decrease during the COVID-19 period. One indicator (5.3%) reported a stable number of diagnostic procedures performed.

#### 3.2.3. Treatment

A total of 221 indicators regarding the ‘treatment’ phase of the cardiac care pathway were identified. These indicators were grouped into three separate categories: ‘procedure numbers’, ‘length of stay,’ and ‘ACS treatment pathway times’.

Procedure numbers

Regarding the number of treatment procedures, 79 indicators were identified from 32 articles [28,30,36,42,43,51,52,53,54,57,58,60,64,65,71,76,78,79,80,82,83,85,88,90,97,98,99,100,101,102,103,104]. The indicators collected mainly concerned the number of percutaneous coronary interventions (Figure 5). Of these indicators, 64 (81%) reported a decrease in the number of treatment procedures performed during the COVID-19 period, and 15 indicators (19%) reported a stable number of procedures.

Length of stay

Regarding the patients’ in-hospital length of stay, 36 indicators were identified from 25 articles [10,29,32,39,43,47,48,50,52,54,57,61,69,71,73,74,75,76,77,80,85,87,88,102,105]. Most of the indicators reported on patients with ACS (Figure 5). Of these indicators, 21 (58%) reported a decrease in the length of stay during the COVID-19 period, 13 indicators (36%) reported a stable length of stay, and 2 indicators (5.6%) reported an increased length of stay.

Acute Coronary Syndrome treatment pathway times

Regarding the treatment times for ACS care, 106 indicators were identified from 35 articles [27,30,33,36,40,41,48,57,61,63,66,69,71,73,74,75,76,77,78,81,82,83,84,85,86,87,88,89,91,92,93,94,95,105,106]. Treatment pathway times reported in the papers were symptom-to-contact, symptom-to-door, symptom-to-diagnosis, symptom-to-balloon, contact-to-door, contact-to-balloon, door-to-ECG, door-to-balloon, ECG-to-balloon, diagnosis-to-balloon, first medical contact-to-catheter laboratory arrival, catheter-to-puncture, catheter laboratory arrival-to-balloon, puncture-to-balloon, and procedure time. Of these indicators, 65 (61%) reported an increase in ACS treatment pathway times during the COVID-19 period, 37 indicators (35%) reported stable treatment pathway times, and 4 indicators (3.8%) reported decreased treatment pathway times.

#### 3.2.4. Outpatient Care

A total of 24 indicators regarding the ‘outpatient care’ phase of the cardiac care pathway were identified. These indicators were grouped into two separate categories: ‘outpatient activity’ and ‘telehealth’.

Outpatient activity volume

Regarding outpatient activity, 18 indicators were identified from eight articles [11,42,53,55,73,107,108,109]. Most of the indicators were concerned only with in-person outpatient activity. Of these indicators, 17 (94%) reported a decrease in outpatient activity volume during the COVID-19 period. One indicator (5.6%) reported a stable number.

Telehealth

Regarding the use of telehealth, six indicators were identified from six articles [11,42,55,107,109,110]. All the indicators reported increased use of telehealth.

#### 3.2.5. Outcomes

A total of 148 indicators regarding the ‘outcomes’ phase of the cardiac care pathway were identified. These indicators were grouped into two categories: ‘outcomes and complications’ and ‘mortality rates’.

Outcomes and complications

Regarding patients’ outcomes and complications, 88 indicators were identified from 31 articles [19,27,30,34,40,43,48,52,59,61,63,64,66,69,74,75,76,77,81,82,83,84,87,88,91,92,93,94,102,103,109]. Most indicators were on patients with ACS, in particular ST-elevated myocardial infarctions (STEMI), such as left ventricular ejection fraction at discharge, thrombolysis in Myocardial Infarction (TIMI) risk score after percutaneous coronary intervention, and major adverse cardiovascular events. Of these indicators, 35 (40%) reported worse outcomes during the COVID-19 period, 46 indicators (52%) reported a stable outcome, and 7 (8.0%) reported a better outcome.

Mortality rates

Regarding mortality rates, 60 indicators were identified from 37 articles [10,19,31,32,33,34,36,37,39,40,41,43,47,48,50,52,54,57,59,60,61,66,69,71,73,74,79,84,87,88,89,90,93,94,102,103,111]. Forty-seven indicators reported on in-hospital mortality rate (Figure 5), from which 22 indicators (47%) signalled an increase, while the same number (22, 47%) reported a stable trend and 3 indicators (6%) signalled a decreasing trend. Six indicators reported on the 30-day death rate (Figure 5), from which five (83%) signalled an increasing trend and one (17%) a stable trend. There were seven indicators reporting changes in mortality rates without any specific interval provided or with a different interval.

Of all the indicators regarding mortality, 31 (52%) reported an increase in mortality rates during the COVID-19 period, 26 (43%) reported stable mortality rates, and 3 indicators (5%) show decreased mortality rates.

An overview of the trends reported by the indicators in each phase of the hospital cardiac care pathway (defined as desirable, undesirable, or stable, following clinical reasoning) is shown in Figure 6. The trends reported for each indicator of the care pathway, by country, are available in Appendix A.

## 4. Discussion

This study aimed to provide an overview of the impact of the COVID-19 pandemic on hospital services for patients with cardiac diseases. We analyzed more than 1600 indicators that were used in 94 papers, reporting on 109 different countries. Our findings show that all phases of the hospital care pathway for patients with cardiac diseases (admission, diagnosis, treatment, outpatient care, and outcomes) were, to different degrees, affected during the pandemic. Admission numbers dropped substantially, and patients arrived later and in a worse clinical condition at the hospital. The number of diagnostic and treatment procedures decreased, ACS treatment pathway times increased, and patients were discharged from the hospital after a shorter length of stay. Outpatient activity decreased, whereas the use of telehealth services increased. Finally, worse clinical outcomes and an increase in mortality rates were reported. Although future crises could lead to different consequences, and even though this study’s results refer to a period without immunization of the population to the SARS-CoV-2 virus, we consider that this information could help to plan future phases of the current pandemic or future events. This study’s results can inform which indicators to monitor closely, support the standardization of indicators, and bolster their embeddedness in health information systems towards enhancing health systems’ resilience and adaptability, allowing to avoid or minimize disruption and postponement of care during crises. This information can inform clinicians and policy makers on the main areas affected in the cardiac care pathway, contributing to the monitoring and improvement of health care delivery. Additionally, these results can be helpful in planning the recovery of care for patients with cardiac diseases.

The drop in admissions to hospital services reported in almost all included articles is probably signaling healthcare avoidance, caused by patients afraid of being infected with SARS-CoV-2 in the hospital [112]. Another explanation could be an actual reduction in the incidence of cardiac diseases during the pandemic. Several explanations have been opted for this reduction in incidence, such as changes in physical activity during lockdowns or a reduction of air pollutants [113,114,115,116]. Patients with cardiac diseases presenting later to the hospital and arriving in a worse clinical condition than before the pandemic could also have been caused by the fear and avoidance mentioned before, with patients waiting for longer periods before seeking care.

Following the drop in admission numbers and decreased outpatient activity, a decrease in the number of diagnostic and treatment procedures was to be expected. On top of patients potentially avoiding care and a possible lower incidence rate of cardiac diseases, elective procedures and surgeries were cancelled [117]. These three factors together have likely contributed to reducing procedure volume. The decreased length of hospital stay that we found in most indicators is explained by several authors by a shortage of hospital bed capacity or the physicians’ intention to minimize the risk of patients’ exposure to the virus. For instance, the European Society of Cardiology recommended that patients “should stay in the hospital for the shortest time possible” [118], and healthcare providers in the United Kingdom were advised to reduce non-ST-elevation myocardial infarction inpatient stays to 36–48 h if normal pathways could not be followed [54]. The increased ACS treatment pathway times and the decreased length of stay together indicate that the hospital cardiac care pathway was under pressure during the COVID-19 pandemic and the delivery of care got squeezed.

The reduction in outpatient care volume that we show in this study mostly comes from indicators reporting on in-patient visits. Therefore, it is probably an underestimation of the actual outpatient care provided. The change towards using telehealth shows that innovative measures were taken during the pandemic to avoid the risk of SARS-CoV-2 infection and health systems adapted quickly to reach out to patients with other means than in-person consultations. In theory, strengthening the use of telehealth to provide continuity of care is beneficial. However, little is known about the quality of care provided and potential inequalities that may have risen from the use of telehealth (e.g., access for all patient groups).

Our findings suggest that patients with cardiac conditions are showing worse outcomes and increased mortality rates during the COVID-19 pandemic. The indicators that reported changes in mortality mostly concerned short-term mortality rates (up to 30 days). An even larger impact on mortality might be found when long-term mortality rates are assessed in later studies. These studies will have to contribute to capturing the real impact that this paper has outlined. The repercussions of care avoidance and cancelled or delayed diagnostic and treatment procedures could reveal themselves in the years to come.

To our knowledge, this is the first scoping review that gives an overview of the impact of the COVID-19 pandemic on the hospital cardiac care pathway in OECD countries. The scoping review methodology gives the benefit of mapping the published literature that became available during the early stages of the COVID-19 pandemic (January–June 2020). While earlier works assessed the impact of the pandemic on specific cardiac diseases care [119], in a specific country [120] or nation-wide, and previous authors have mentioned some of the impact revealed in this work based on earlier articles [121], this study has the strength to present an overview of performance indicators’ trends collated in a systematic way, which capture the impact throughout the hospital cardiac care pathway with an international scope. This methodology, while it constitutes an innovative approach, allowed to map performance indicators that could be used by countries to assess the impact of the pandemic as it evolves, in a more uniform and comparable way.

However, a scoping review has its limitations. Despite efforts to accelerate the time of scientific publication, there is still a relevant time lag between the occurrence of events and their reporting. The time it takes to perform a review related to a broad clinical area justifies why this study only reports results from the first semester of 2020. Nonetheless, our findings provide a thorough overview of performance indicators across the cardiac care pathway, which can contribute to better monitoring the quality of cardiac care provided. Ideally, a systematic review with meta-analyses would be performed. For the articles included in this paper, this would be extremely difficult, given their different methodologies, indicators, indicators’ definitions, and comparison periods. The heterogeneity in study characteristics and indicators make it hard to compare data in a reliable way. In July 2020, the International Training Network for Healthcare Performance Intelligence Professionals (HealthPros) [122] suggested that to compare, manage, and improve health systems responsiveness to the pandemic, commonly agreed-upon standardized data and indicators are necessary [123]. We recommend devising a uniformly accepted set of indicators with clear definitions to use in future pandemics.

Another limitation is that only articles reporting on OECD countries were included in this scoping review. The global repercussions of the COVID-19 pandemic on hospital services for cardiac care will therefore likely be larger than reported in this paper, given the smaller capacity to handle changes in hospital care in low-income countries. On the other hand, publication bias may have played a role in portraying a more severe impact of the pandemic. Physicians might publish sooner when the effects of the pandemic are being clearly noticed in their hospital. It could also be the case that the overload with clinical tasks did not allow physicians to find the time to do scientific work, which may counterbalance this limitation. Regardless, we consider our findings to be useful and signaling major trends.

Based on the included articles, we are not able to provide an analysis of what caused these changes in the hospital cardiac care pathway. Being able to contextualize these results with future research will be of use for implementing measures to improve care during the current and future pandemics or disasters. Studies systematically assessing the following phases of the pandemic are necessary to evaluate whether the cardiac patients’ management improved. We also welcome studies on the quality and access of telemedicine. It would be relevant to study if and how this could improve quality and access of services delivery for the better, also in non-crisis times.

## 5. Conclusions

These results signal that the hospital services delivery process for cardiac care came under pressure in the first half of 2020 and all phases of the hospital cardiac care pathway were affected. Lessons should be learnt and steps taken to be able to safeguard the continuity of care during the ongoing COVID-19 pandemic and in future crises. Furthermore, to guide the decisions of health system actors on the implementation of measures to ensure the continuation of essential care during future crises, fostering the use of an international standardized set of indicators is paramount, making optimal use of existing health information infrastructures.

## Figures and Tables

**Figure 1 ijerph-19-03172-f001:**
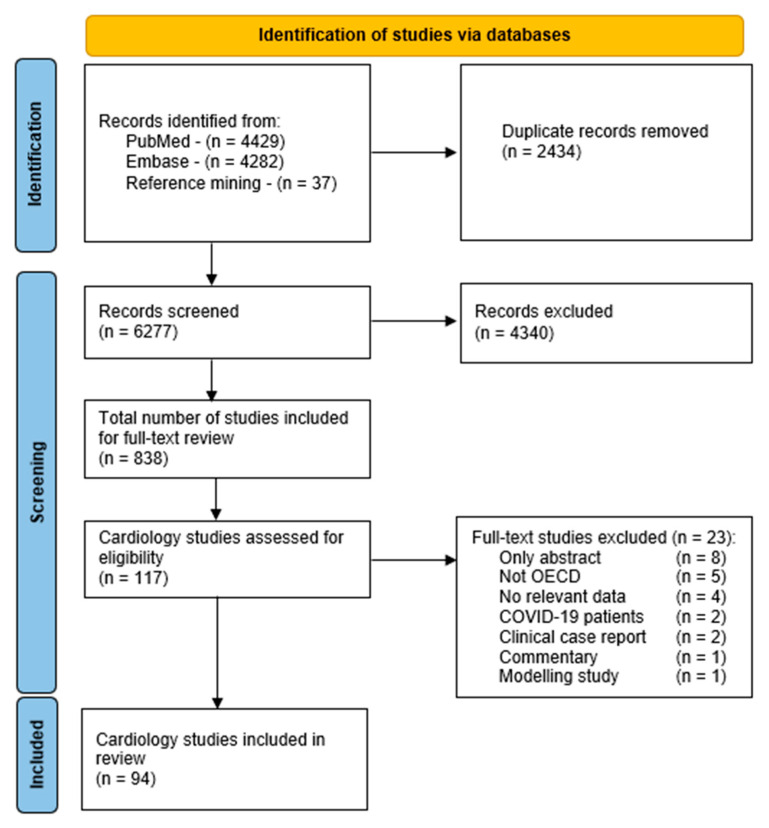
PRISMA flow diagram of the literature search. Abbreviations: OECD—Organization for Economic Co-operation and Development.

**Figure 2 ijerph-19-03172-f002:**
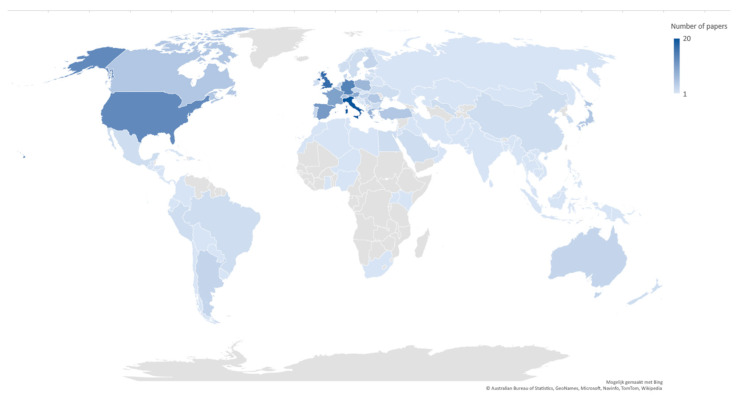
Countries reported on in the included articles (including articles reporting on multiple countries), color-graded according to the number of included papers (n = 109).

**Figure 3 ijerph-19-03172-f003:**
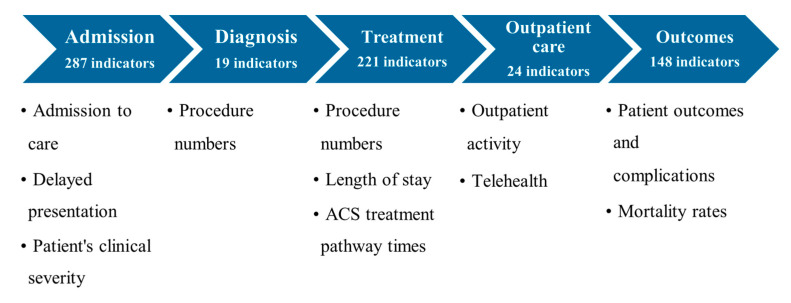
Categorization of indicators according to different phases of the hospital cardiac care pathway. Abbreviations: ACS—Acute Coronary Syndrome.

**Figure 4 ijerph-19-03172-f004:**
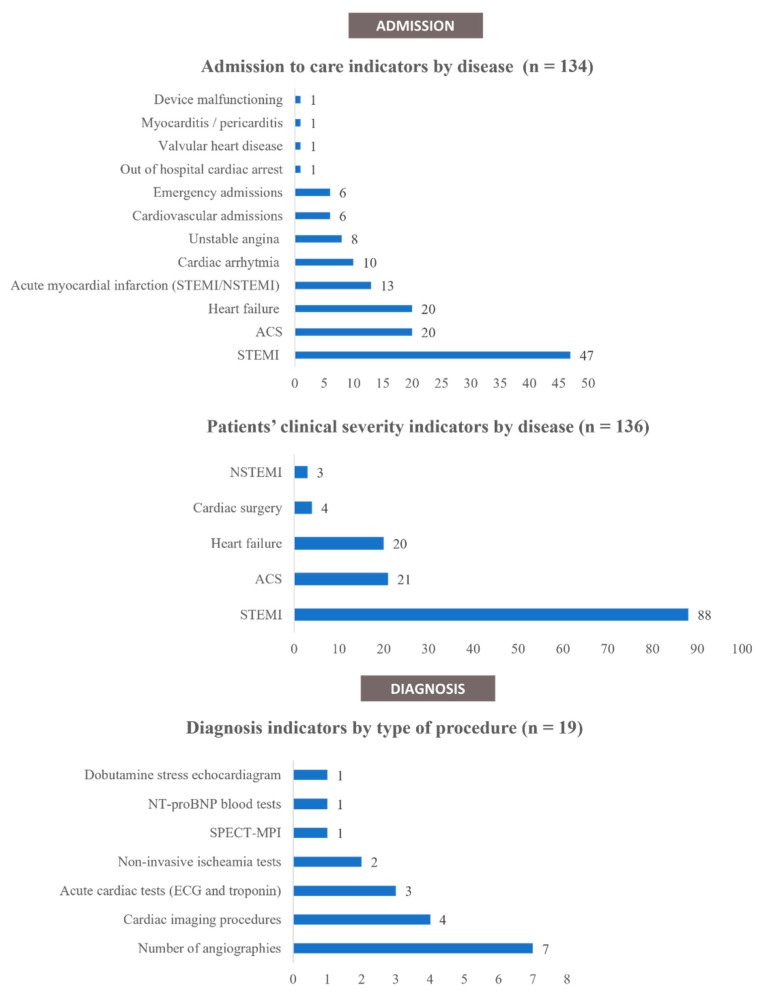
Admission and diagnosis indicators by disease. Abbreviations: ACS—Acute coronary syndromes; NSTEMI—Non-ST-elevation myocardial infarctions; SPECT-MPI—Single-photon emission computed tomography myocardial perfusion imaging; STEMI—ST-elevated myocardial infarctions.

**Figure 5 ijerph-19-03172-f005:**
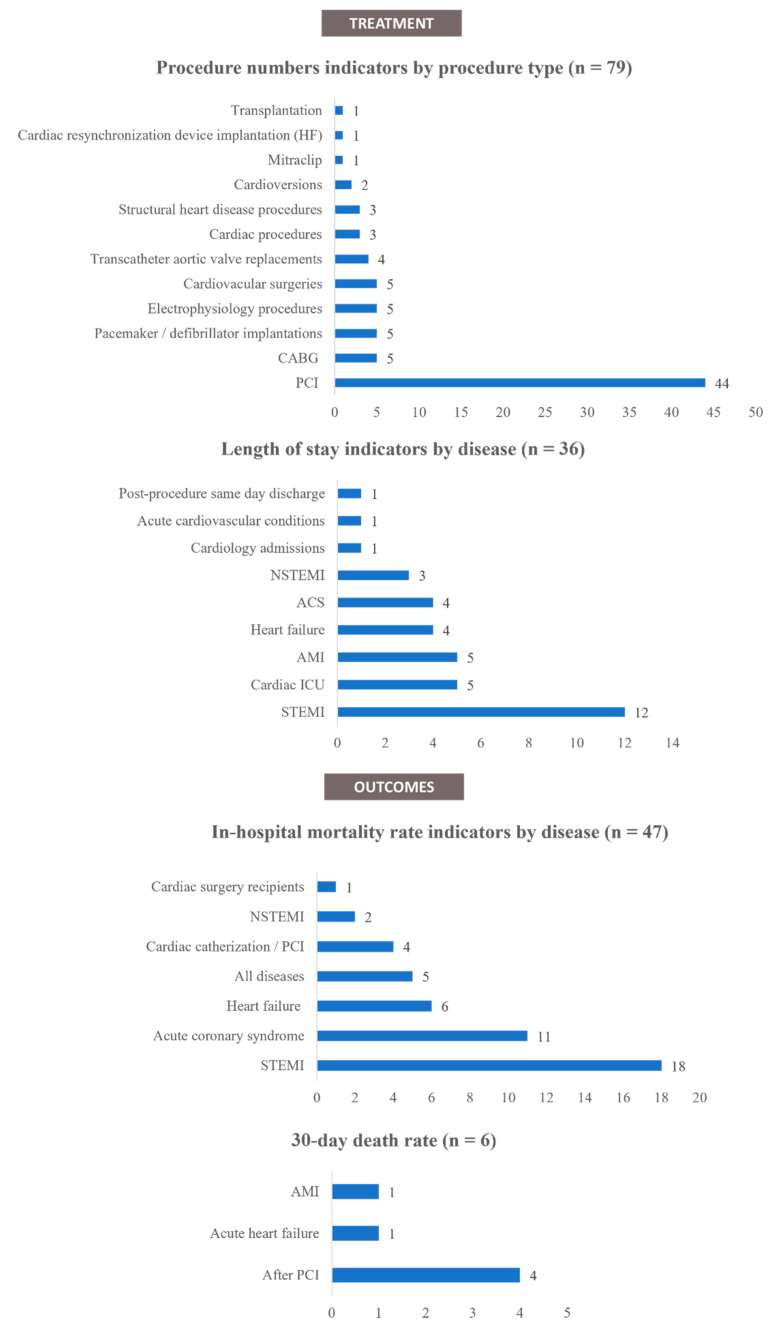
Treatment and outcome indicators by disease. Abbreviations: ACS—Acute coronary syndromes; AMI—Acute myocardial infarction; CABG—coronary artery bypass graft; HF—Heart failure; ICU—Intensive care unit; NSTEMI—Non-ST-elevation myocardial infarctions; PCI—Percutaneous coronary intervention; STEMI—ST-elevated myocardial infarctions; SPECT-MPI—Single-photon emission computed tomography myocardial perfusion imaging.

**Figure 6 ijerph-19-03172-f006:**
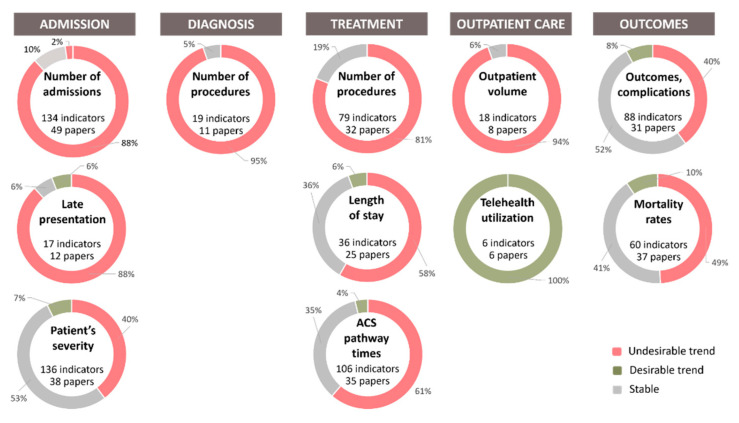
Hospital Cardiac Care Pathway Indicators’ Trends during the COVID-19 Pandemic’s early stages (January–June 2020).

## Data Availability

Data underlying this article is available in Zenodo.org, at https://dx.doi.org/10.5281/zenodo.5745755, accessed on 28 January 2022.

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
