# Peer review of "The Impact of the COVID-19 Pandemic on Hospital Services for Patients with Cardiac Diseases: A Scoping Review"

_ijerph, 2022, doi:10.3390/ijerph19063172_

Round 1
Reviewer 1 Report
I have read the publication submitted for review with great interest. The work is a systematic review of the available literature in which an assessment of the impact of the COVID-19 pandemic on cardiac treatment was undertaken. Due to the nature of the work, the authors of which have undertaken to sum up the above-mentioned influence, the reviewer may only refer to the technique of the prepared work, which seems to be completely correct. The introduction introduces the topic, the results are clearly discussed. Of particular importance is the figure 4 showing the desired and undesirable trends in the steps of therapeutic interventions in cardiology. Unfortunately, it is clear that undesirable trends are dominant, which translates into an increase in "health debt" in cardiology.
Due to the nature of the work and the results, I believe that the work is a uniform whole, it can be published in the current form. I also believe that the publication is a valuable study as an argument in discussions with the payer (e.g. health funds or insurers) regarding the need to increase financing in cardiology.
Element that could be improved: I believe that the work (main manuscript) lacks information on what cardiovascular diseases and what hospitalizations are the subject of the study. For example, in many cardiology departments, patients with trauma, after surgery, orthopedics, gynecology, sepsis, and cancer are hospitalized due to the lack of places in dedicated departments. This clearly affects the profile and statistical data derived from units on assessment times at work. These data can be found in additional materials, but I would add one figure characterizing the type of hospitalization (saving life, urgent, elective), assessed diagnoses, etc., so that in the main part of the work, I would allow for better reaching the conclusions made.
Reviewer 2 Report
I’ve read with attention the paper of de Lange et al. that is potentially of interest. The background and aim of the study have been clearly defined. The methodology applied is overall correct, the results are reliable and adequately discussed. I’ve only some minor comments. In particular, the authors conclude that this information could support the planning of care during the ongoing pandemic and in future events. This could not be true, because the data were sampled were a large part of the population was not vaccinated nor immunized, and because we can't foresee the characteristics of an eventual new pandemia related to a different COVID-variant. This has to be considered and commented. The authors should also comment on why the report a review related to the period January-June 2020, while we are at the beginning of 2022, and a lot of paper on this argument have been published in the second half of 2020 and during the whole 2021.
